# Autoimmune Diseases and COVID-19 as Risk Factors for Poor Outcomes: Data on 13,940 Hospitalized Patients from the Spanish Nationwide SEMI-COVID-19 Registry

**DOI:** 10.3390/jcm10091844

**Published:** 2021-04-23

**Authors:** María del Mar Ayala Gutiérrez, Manuel Rubio-Rivas, Carlos Romero Gómez, Abelardo Montero Sáez, Iván Pérez de Pedro, Narcís Homs, Blanca Ayuso García, Carmen Cuenca Carvajal, Francisco Arnalich Fernández, José Luis Beato Pérez, Juan Antonio Vargas Núñez, Laura Letona Giménez, Carmen Suárez Fernández, Manuel Méndez Bailón, Carlota Tuñón de Almeida, Julio González Moraleja, Mayte de Guzmán García-Monge, Cristina Helguera Amezua, María del Pilar Fidalgo Montero, Vicente Giner Galvañ, Ricardo Gil Sánchez, Jorge Collado Sáenz, Ramon Boixeda, José Manuel Ramos Rincón, Ricardo Gómez Huelgas

**Affiliations:** 1Internal Medicine Department, Regional University Hospital of Málaga, Biomedical Research Institute of Málaga (IBIMA), University of Málaga (UMA), 29010 Málaga, Spain; carlosrg1968@gmail.com (C.R.G.); ivanpdp@hotmail.com (I.P.d.P.); ricardogomezhuelgas@hotmail.com (R.G.H.); 2Internal Medicine Department, Bellvitge University Hospital-IDIBELL, L’Hospitalet de Llobregat, 08907 Barcelona, Spain; mrubio@bellvitgehospital.cat (M.R.-R.); amontero@gencat.cat (A.M.S.); nhoms@bellvitgehospital.cat (N.H.); 3Internal Medicine Department, 12 de Octubre University Hospital, 28041 Madrid, Spain; blanca.ayuso90@gmail.com; 4Internal Medicine Department, Gregorio Marañon University Hospital, 28007 Madrid, Spain; mariacarmen.cuenca@salud.madrid.org; 5Internal Medicine Department, La Paz University Hospital, 28046 Madrid, Spain; farnalich@salud.madrid.org; 6Internal Medicine Department, Albacete University Hospital Complex, 02006 Albacete, Spain; jlbeato@sescam.org; 7Internal Medicine Department, Puerta de Hierro University Hospital, 28222 Majadahonda, Spain; juanantonio.vargas@uam.es; 8Internal Medicine Department, Miguel Servet Hospital, 50009 Zaragoza, Spain; laura.letona.g@gmail.com; 9Internal Medicine Department, La Princesa University Hospital, 28006 Madrid, Spain; csuarezfe@gmail.com; 10Internal Medicine Department, San Carlos Clinical Hospital, 28040 Madrid, Spain; manuelmenba@hotmail.com; 11Internal Medicine Department, Zamora Hospital Complex, 49022 Zamora, Spain; carlottadealmeida@gmail.com; 12Internal Medicine Department, Virgen de la Salud Hospital, 45004 Toledo, Spain; juliogmoraleja@gmail.com; 13Internal Medicine Department, Infanta Cristina University Hospital, 28981 Parla, Spain; maytedeguzman@gmail.com; 14Internal Medicine Department, Cabueñes Hospital, 33394 Gijón, Spain; cristina.h.amezua@gmail.com; 15Internal Medicine Department, Henares Hospital, 28822 Coslada, Spain; garrotefidalgo@telefonica.net; 16Internal Medicine Department, San Juan de Alicante University Hospital, 03550 San Juan de Alicante, Spain; ginervicgal@gmail.com; 17Internal Medicine Department, La Fe University Hospital, 46026 Valencia, Spain; rigilsan@gmail.com; 18Internal Medicine Department, San Pedro Hospital, 26006 Logroño, Spain; jorgecolladosaenz@hotmail.com; 19Internal Medicine Department, Mataró Hospital, 08304e Mataró, Spain; Rboixeda@ub.edu; 20Department of Clinical Medicine, Miguel Hernandez University of Elche, 03202 Alicante, Spain; jramosrincon@yahoo.es

**Keywords:** autoimmune diseases, antirheumatic agents, biological therapy, glucocorticoids, immune system diseases, COVID-19, SARS-CoV-2

## Abstract

(1) Objectives: To describe the clinical characteristics and clinical course of hospitalized patients with COVID-19 and autoimmune diseases (ADs) compared to the general population. (2) Methods: We used information available in the nationwide Spanish SEMI-COVID-19 Registry, which retrospectively compiles data from the first admission of adult patients with COVID-19. We selected all patients with ADs included in the registry and compared them to the remaining patients. The primary outcome was all-cause mortality during admission, readmission, and subsequent admissions, and secondary outcomes were a composite outcome including the need for intensive care unit (ICU) admission, invasive and non-invasive mechanical ventilation (MV), or death, as well as in-hospital complications. (3) Results: A total of 13,940 patients diagnosed with COVID-19 were included, of which 362 (2.6%) had an AD. Patients with ADs were older, more likely to be female, and had greater comorbidity. On the multivariate logistic regression analysis, which involved the inverse propensity score weighting method, AD as a whole was not associated with an increased risk of any of the outcome variables. Habitual treatment with corticosteroids (CSs), age, Barthel Index score, and comorbidity were associated with poor outcomes. Biological disease-modifying anti-rheumatic drugs (bDMARDs) were associated with a decrease in mortality in patients with AD. (4) Conclusions: The analysis of the SEMI-COVID-19 Registry shows that ADs do not lead to a different prognosis, measured by mortality, complications, or the composite outcome. Considered individually, it seems that some diseases entail a different prognosis than that of the general population. Immunosuppressive/immunoregulatory treatments (IST) prior to admission had variable effects.

## 1. Introduction

The novel coronavirus 2019 disease (COVID-19) pandemic has spread throughout the world. Spain is one of the most affected countries in Europe, with more than 3,153,000 cumulative cases [1].

The infection has a more severe course in elderly men or those with comorbidities [2,3]. In the case of patients with autoimmune diseases (ADs), questions have been raised about the true susceptibility to SARS-CoV-2 infection and the course of the AD after infection. In general, the underlying dysfunction of their immune system and the immunosuppressive/immunoregulatory treatments (ISTs) they receive mean that some patients with ADs are at greater risk of infection [4,5]. Different studies have observed a frequency of COVID-19 in these patients that is higher than that of the general population [6,7,8], as well as a worse prognosis in terms of the need for mechanical ventilation (MV) or admission to intensive care units (ICUs) [9]. In others, however, the frequency of COVID-19 [10,11,12,13,14] and its prognosis were similar to patients without ADs [13,15,16,17,18,19]. It has been suggested that patients with ADs could have adopted stricter and earlier self-protection and social distancing measures than the general population [12,20].

On the other hand, antimalarials, which are common in the treatment of ADs, have in vitro antiviral effects [21], although no protective effect has been demonstrated [9,12,22,23]. In the inflammatory phase of COVID-19, ISTs could protect patients with ADs who usually receive them from the onset of a cytokine storm, which is associated with a poor prognosis [14,15,24]. Finally, some ADs have basal characteristics that could influence the course of COVID-19, both positively (overexpression of interferon alpha [25]) and negatively (lymphopenia [26] or epigenetic dysregulation [27]).

The Spanish Society of Internal Medicine (SEMI) has created a registry of patients hospitalized due to COVID-19 in Spain. This registry makes it possible to compare the sociodemographic and clinical characteristics and the clinical course of patients with and without ADs in order to establish whether differences exist between them.

## 2. Materials and Methods

### 2.1. Study Design and Patients

We have used the information available in the Spanish nationwide SEMI-COVID-19 Registry, which retrospectively compiles data from the first admission of patients ≥ 18 years of age with COVID-19, confirmed microbiologically by a reverse transcription polymerase chain reaction (RT-PCR) test of a nasopharyngeal swab sample, sputum specimen, or bronchoalveolar lavage. This registry collects sociodemographic data, previous medical history, routine treatment (including CS, antimalarials, conventional or targeted disease-modifying anti-rheumatic drugs (cs/tsDMARDs), and bDMARDs), clinical presentation, clinical condition (including the degree of functional dependence, as evaluated by the Barthel Index [28], and the presence of comorbidities, as evaluated by the Charlson Comorbidity Index [29]), laboratory test results, radiological findings, clinical management, in-hospital complications, length of hospital stay, early readmissions, referral to long-term care or skilled nursing facilities, and deaths. Patients were treated at their attending physician’s discretion according to local protocols and their clinical judgment.

The SEMI-COVID-19 Registry was approved by the Provincial Research Ethics Committee of Málaga (Spain). Given that it is observational in nature, the registry poses no additional inconvenience to the patients included. Informed consent was obtained from all patients. When it was not possible to obtain it in writing for biosecurity reasons or because the patient had already been discharged, it was collected verbally and noted on the medical record. More in-depth information and preliminary results of the SEMI-COVID-19 Registry have recently been published [30].

We selected all of the patients with ADs included in the registry until 30 June 2020 (prior to the beginning of the vaccination campaign in Spain) and compared them with the remaining patients. The ADs were defined based on the information included in the medical record and the judgment of the physician in charge of entering the data into the registry, which was done using a standardized online data capture system. The primary outcome of the study was all-cause mortality during admission, readmission, and subsequent admissions. Secondary outcomes were a composite outcome, including the need for ICU admission, invasive and non-invasive MV, or death, as well as in-hospital complications. The latter included secondary bacterial pneumonia, acute respiratory distress syndrome (ARDS), acute heart failure, arrhythmia, acute coronary syndrome, myocarditis, seizures, stroke, shock, sepsis, acute kidney failure, disseminated intravascular coagulation (DIC), venous thromboembolism, multiple organ dysfunction syndrome, and acute limb ischemia.

### 2.2. Statistical Analysis

Categorical variables were presented as a number (percentage), and continuous variables were reported as the mean ± standard deviation (SD). Categorical variables were compared using the chi-squared or Fisher’s exact tests. Continuous variables were compared using an ANOVA test. The level of significance was established as a two-tailed *p* < 0.05. No corrections were made for multiple comparisons.

The different models of logistic regression were developed with the group of patients in the registry without ADs who had valid information in all of the predictor and result variables included in the corresponding analysis. For patients with ADs, the missing data were completed by multiple imputations [31]. Multivariable logistic regression was used to estimate the odds ratio (OR) and 95% confidence interval (95% CI) when comparing outcomes, mortality, composite outcomes, and complications during hospitalization. The regression model included sociodemographic variables, comorbidities, and prior ISTs. For the predictor variable selection process, the Wald statistic, forward method, was used, with inclusion *p* < 0.05 and exclusion *p* < 0.10.

As it is an observational, non-randomized study, to reduce the number of model predictor variables, avoid selection biases, and better control the influence of their possible confounding effect, the different propensity scores (PSs) were independently calculated [32,33] for the binary variables of ADs, systemic lupus erythematosus (SLE), rheumatoid arthritis (RA), primary Sjögren syndrome (PSS), systemic sclerosis (SSc), mixed connective tissue disease (MCTD)/overlap syndrome, inflammatory myopathies (IM), primary antiphospholipid syndrome (APS), spondyloarthropaties, vasculitis (systemic vasculitis, including giant cell arteritis), polymyalgia rheumatica (PMR), and combined PMR/giant cell arteritis.

In the first step, in the logistic regression model that included the previously cited variables as dependents and variables on sociodemographic data, comorbidity, preadmission ISTs, and drugs received during the hospital stay as predictors, the estimated probability for each dependent variable was calculated as a PS using the enter method. In the next step, this PS was weighted by calculating its inverse (inverse propensity score weighting (IPSW) method) in patients with AD as 1/PS and in patients without AD as 1/(1-PS); the histogram of the weighted scores showed that the groups were comparable. Subsequently, an analysis of generalized estimation equations was carried out in the generalized linear models module of the SPSS statistical package in order to retrieve the original sample sizes and calculate the OR with their 95% CI. To assess the robustness of the results, sensitivity analyses were performed, comparing the results of the logistic regression analysis with those obtained through the IPSW method. All analyses were conducted using IBM SPSS Statistics for Windows, Version 22.0. (Armonk, NY: IBM Corp., US).

## 3. Results

### 3.1. Patients

As of 30 June 2020, a total of 13,940 patients diagnosed with COVID-19 were included in the SEMI-COVID-19 Registry, of which 362 (2.6%) had ADs, which were sub-classified into classic ADs, other ADs, and miscellaneous ADs (Table 1).

The average age of patients with ADs was somewhat higher, and female sex was more frequent than in the general population. They had greater comorbidity, with a Charlson Index > 2, especially in the group of patients with classic ADs (2.66 vs. 1.27 in the general population). The clinical manifestations, physical examination on admission, and radiological alterations of patients with ADs were similar to the general population (Table 2 and Appendix A). Patients with ADs were being treated with ISTs more frequently (Table 2).

During admission, patients with AD received ISTs more frequently than the general population, especially CS and immunoglobulins (IGIV), while the use of antimalarial drugs, which were part of the standard of care during a certain period of time in Spain, was similar in all groups (Table 3).

### 3.2. Prognosis in Patients with ADs

Although the duration of admission was similar in all groups, in the univariate analysis, patients with ADs had more complications (*p* = 0.008) and higher mortality (*p* < 0.001), and met the criteria of the composite outcome more often than the general population (*p* < 0.001), especially the group of classic ADs and other ADs (Table 4).

In multivariate logistic regression analysis, the ADs as a whole showed neutral effects on the outcome variables (mortality, composite outcome, and complications). Habitual treatment with CS was associated with a greater risk of these outcome variables, unlike the rest of the habitual ISTs, which had a neutral effect. Age, male sex, Hispanic ethnicity/race, Barthel Index score, and comorbidity (Charlson Comorbidity Index, hypertension, dyslipidemia, obesity, respiratory pathology, non-arteriosclerotic cardiovascular disease, and transplantation) were also poor prognostic factors (Table 5).

When we analyzed the effect of ADs as a whole and individually, as well as the habitual ISTs on the outcome variables by calculating the IPSW method, ADs as a whole were not associated with higher mortality, increased risk of the composite outcome, or more complications during admission (Table 6). Some Ads, such as RA, PMR, and vasculitis, were associated with higher mortality; spondyloarthropathies, vasculitis, and PMR were associated with a higher risk of the composite outcome, as well as RA, with higher risk of complications during hospitalization. SSc was associated with a lower risk of mortality, composite outcome, and complications during hospitalization. Similarly, PSS was associated with fewer complications during hospitalization (Appendix A). The effect of the habitual ISTs was very variable. BDMARDs were associated with a decrease in mortality in the set of Ads, and were especially beneficial in spondyloarthropathies, although contradictory results were observed in SLE, RA, and vasculitis. CS did not affect the overall ADs, but was detrimental in some diseases, such as PSS, SSc, MCTD/overlap, APS, IM, and vasculitis. Cs/tsDMARDs also did not affect the overall group, but were beneficial in some diseases, such as PMR/giant cell arteritis, SLE, and RA. Finally, antimalarials did not change the prognosis.

## 4. Discussion

In this study, we analyzed data from one of the largest cohorts of hospitalized patients with ADs and COVID-19 published to date. In this registry, patients with ADs accounted for 2.6% of the total number of patients admitted to 150 hospitals in Spain for COVID-19. Other authors have described figures that range from ≤ 1% to 10% [34,35,36]. This variability could be influenced by the different prevalence of ADs according to the geographical area [37], as well as the selection and classification criteria of the ADs used. In the nationwide Spanish SEMI-COVID-19 Registry, several ADs were included based on the dysfunction of the underlying immune system, the possibility of receiving ISTs, and the judgment of the clinician in charge of data entry, which were gathered from the information contained in the medical record. The actual incidence of COVID-19 in patients with ADs is still unknown. Worldwide, in different epidemiological contexts and for different ADs, its incidence has been described as being both similar [10,12,13,14,38,39] and higher [6,7,40] than among the general population. In our case, the design of the registry does not allow for an approximation in this regard beyond what has been observed in hospitalized patients in our country.

In this registry, patients with ADs were more frequently female, older, more dependent, and with greater comorbidity than patients without ADs. In the univariate analysis, patients with ADs had higher mortality, met the criteria of the composite outcome more often, and had more complications during hospitalization. However, after controlling for the effect of possible known confounding factors using multivariate logistic regression and the IPSW method, the presence of an ADs did not entail a greater risk of these variables.

Two international registries of patients with ADs and a recent meta-analysis have shown the deleterious effect of age and comorbidity on the prognosis of COVID-19 [8,41,42,43]. These same factors may have acted as confounders of the higher mortality, composite outcome, and complications that we observed in the univariate analysis in patients with ADs. In the multivariate analysis, in which the effect of ADs on the prognosis of COVID-19 was estimated, taking into account these and other confounding factors (age, sex, ethnicity/race, comorbidities, alcohol and tobacco use, degree of dependency, and ISTs before and during admission), no relationship with the prognosis was observed. Age, male sex, and comorbidity behaved as poor prognostic factors in patients with ADs, as well as in the general population. Different studies published around the world support the idea that patients with ADs and COVID-19 have a similar clinical course to that of the general population [6,12,13,17,34,35,44,45]. Only two cohorts, both of which included only a small number of patients with ADs, found a higher frequency of respiratory failure [46] and need for MV or ICU admission [9] in patients with ADs compared to the general population, although there were no differences in the mortality rates in both cases.

Few studies have assessed the effect of different ADs on the prognosis of COVID-19. In our study, analyzed individually, worse progress was observed in patients with certain ADs, including RA, PMR, vasculitis, and spondyloarthropathies, and better progress in others, such as PSS and SSc (Appendix A). Other authors have found a greater risk of severe COVID-19 in patients with systemic vasculitis, SLE, PSS, and APS, but they did not find differences between the control group and the group of patients with chronic inflammatory arthritis [47]. Although, in general, having an AD does not necessarily imply a worse prognosis, it cannot be ruled out that specific diseases with different pathogenic mechanisms and phenotypic characteristics would entail a different clinical course than the rest. Until now, the small number of patients with specific diseases that have been included in the publications, the absence of a control group, and the lack of control of biases and confounding factors have not allowed for the effects of these diseases to be fully characterized. Further analysis would be needed to evaluate the severity of specific conditions within the heterogeneous AD group.

Another concern during the pandemic has been the impact of basal ISTs (CS, antimalarials, and DMARDs) on the prognosis of COVID-19. The onset of hyper-inflammatory syndromes and even immune reconstitution inflammatory syndrome associated with poor prognosis have been described in the course of SARS-CoV-2 infection [48]. Immunosuppressive therapies have been used for the treatment of this inflammatory phase of the disease [24,49], similar to other ADs associated with viral infections (such as cryoglobulinemia and hepatitis C virus or polyarteritis nodosa and hepatitis B virus [50]). In addition, ensuring that ADs remain in remission helps prevent infections in general [51] or the least poor progression of them [52]. In this context, various authors have suggested that basal ISTs would not result in a more severe clinical course [45], and could even have some protective effect [14,15,21,49]. In this study’s multivariate logistic regression analysis, the habitual treatment with CS was associated with higher mortality among patients with ADs. In the analysis, by calculating the PS, habitual CS did not influence the prognosis of ADs as a whole, although we did find a significant influence in a few diseases. Two large registries of patients with ADs and a recent meta-analysis have also shown the deleterious effect of CS on the prognosis of COVID-19 [8,41,42,43]. No information was collected in this registry on the dose or duration of pre-admission CS treatment or the activity of the underlying disease, which may have influenced these results.

Regular treatment with antimalarials did not have any protective effect, which is consistent with the results of other observational studies in patients with ADs [9,12,23,41]. However, in patients with PMR, antimalarials were associated with more complications, although the 95% CI was quite wide, and thus was too imprecise to be taken into consideration.

In our study, the regular use of DMARDs (bDMARDs or cs/tsDMARDs) tended to have a neutral or protective effect against the outcome variables in the whole of the ADs and in specific diseases. Other authors have also found a neutral [6,16,38,42,47,53] or protective [41] effect of bDMARDs and cs/tsDMARDs [41,47,53]. Better control of disease activity, as well as the attenuation or prevention of an excessive inflammatory response to COVID-19, could explain the protective or neutral effect of these drugs. In the case of SLE and RA, however, bDMARDs were shown to be harmful. A lack of information about other possible confounding factors, such as AD activity, may have influenced this result.

One of the main strengths of our study is the large number of patients included from the internal medicine departments of more than 150 hospitals. Furthermore, data were collected using a standardized, previously agreed upon protocol. Multiple imputations of the missing data, both randomized and biased, were performed on patients with ADs, which allowed all of them to be analyzed. To avoid selection biases and control for confounding factors in the estimation of the effect of ADs, a multivariate analysis was carried out, and the PS was estimated using the IPSW method. Finally, as a sensitivity analysis, the results obtained in the previous analyses were compared.

Our study has some limitations. In addition to being retrospective, the registry was not specifically designed to evaluate the effect of ADs, but rather was designed as a general registry of patients hospitalized due to COVID-19. This registry took advantage of the information available in electronic medical records, but the data were collected by a large number of researchers from different centers who were not necessarily specialized in ADs or in charge of patient care. This could have led to heterogeneity in the collection and validation of this information. Patients who were unable to provide informed consent at any time, either because of severe illness or mechanical ventilation, were excluded. However, they represent an extremely small proportion of patients, and thus have little impact on the results. The group of patients classified as having ADs included patients with various diseases. Their inclusion in this group was based on the baseline dysfunction of the immune system, the possibility of receiving immunosuppressive treatment, and the judgment of the physician in charge of entering the data, which reflects the habitual clinical practice and allowed us to increase the sample size. Since it was a very heterogeneous group, a subgroup analysis (classical, other, and miscellaneous) and analyses of specific entities were also performed. In the analysis, general population cases that were missing data for any of the variables of the different statistical analyses carried out were excluded from the control group, but, given the large sample size, the power of the statistical contrasts was not affected. On the other hand, the effect of the AD activity or the antiviral or antimicrobial drugs used during hospitalization were not evaluated, although some of the latter have not been proven effective in other studies [54]. In order to create the multivariate statistical contrast models, the admission date was not included. This may entail a selection bias, since, although little time has passed since the beginning of the pandemic, the management of patients may have changed in these months.

## 5. Conclusions

In conclusion, this analysis of the SEMI-COVID-19 Registry found that ADs do not entail a different prognosis, as measured by mortality, the composite outcome (mortality/ICU admission/need for MV), or complications, than that of the rest of patients hospitalized due to COVID-19 in Spain. Considered individually, it cannot be ruled out that some diseases entail a better or worse prognosis than the general patient population. The use of ISTs before admission had variable effects.

More studies are needed, including outpatient cohort studies, in order to determine the different aspects that affect COVID-19 prognosis in patients with ADs.

## Figures and Tables

**Table 1 jcm-10-01844-t001:** Classification of the autoimmune diseases (ADs).

	Disease	*n* (%)	Total (%)
Without AD		13,578 (97.4)	13,578 (97.4)
Classical AD	RA	113 (0.81)	207 (1.48)
	SLE	23 (0.16)	
	PSS	19 (0.14)	
	SSc	13 (0.09)	
	Vasculitis: ANCA-vasculitis (8), isolated CNS vasculitis, Schönlein- Henoch purpura, urticarial hypocomplementemic vasculitis, erythema elevatum with vasculitis, antiMPO/antiGBM glomerulonephritis	13 (0.09)	
	Giant cell arteritis	9 (0.06)	
	MTCD/overlap	9 (0.06)	
	IM	4 (0.03)	
	APS	4 (0.03)	
Other AD	PMR	48 (0.34)	134 (0.96)
	Spondyloarthropathy	33 (0.24)	
	Sarcoidosis	10 (0.07)	
	Inflammatory bowel disease	10 (0.07)	
	Idiopathic pulmonary fibrosis	9 (0.06)	
	Primary biliary cirrhosis	6 (0.04)	
	Autoimmune cytopenia: immune thrombocytopenic purpura (3), autoimmune hemolytic anemia, autoimmune neutropenia	5 (0.04)	
	Autoimmune hepatitis	4 (0.03)	
	Myasthenia Gravis	4 (0.03)	
	Behçet’s disease	3 (0.02)	
	Multiple sclerosis	2 (0.01)	
Miscellaneous AD	Psoriasis	6 (0.04)	21 (0.15)
	Cutaneous lupus	4 (0.03)	
	Seronegative polyarthritis	3 (0.02)	
	Seronegative arthritis	1 (0.01)	
	Arthralgia with antinuclear/citrullinatedpeptide antibody	1 (0.01)	
	Anterior ischemic optic neuropathy/orbital pseudotumor	1 (0.01)	
	Polyglandular autoimmune syndrome	1 (0.01)	
	Collagenous colitis	1 (0.01)	
	Erythema nodosum	1 (0.01)	
	HLAB27 uveitis	1 (0.01)	
	Palindromic rheumatism	1 (0.01)	
Total			13,940 (100)

AD: autoimmune diseases; AntiGBM: anti-glomerular basement membrane antibodies; AntiMPO: myeloperoxidase antibodies; APS: primary antiphospholipid syndrome; CNS: central nervous system; IM: inflammatory myopathies; MTCD: mixed connective tissue disease; PMR: polymyalgia rheumatica; PSS: primary Sjögren syndrome; RA: rheumatoid arthritis; SLE: systemic lupus erythematosus; SSc: systemic sclerosis.

**Table 2 jcm-10-01844-t002:** Baseline characteristics among groups.

	Without AD	Classical AD	Other AD	Miscellaneous AD	*p*-Value
Sociodemographic data					
Age, years (mean ± SD)	67.2 ± 16.3	69.6 ± 13.6	71.3 ± 14.3	66.4 ± 10.8	0.006
Sex (female) *n* (%)	5784 (42.6)	124 (59.9)	64 (47.8)	10 (47.6)	<0.001
Race n (%)					0.040
- Caucasian - Black - Hispanic - Asian - Other	11,997 (89.7) 51 (0.4) 1145 (8.6) 60 (0.4) 118 (0.9)	193 (93.2) 3 (1.4) 10 (4.8) 0 (0.0) 1 (0.5)	129 (96.3) 0 (0.0) 3 (2.2) 0 (0.0) 2 (1.5)	21 (100) 0 (0.0) 0 (0.0) 0 (0.0) 0 (0.0)	
Degree of dependence (Barthel Index score) *n* (%)					0.081
- Absent/mild - Moderate - Severe	11,170 (83.5) 1249 (9.3) 962 (7.2)	166 (80.2) 28 (13.5) 13 (6.3)	105 (78.4) 13 (9.7) 16 (11.9)	20 (95.2) 1 (4.8) 0 (0.0)	
Comorbidities					
Charlson Comorbidity Index (mean ± SD)	1.27 ± 1.80	2.66 ± 2.09	2.36 ± 2.11	2.05 ± 1.66	<0.001
Smoking *n* (%)					0.068
- Non smoker - Smoker	12,228 (94.6) 696 (5.4)	203 (98.1) 4 (1.9)	129 (96.3) 5 (3.7)	21 (100) 0 (0.0)	
Alcohol use disorder *n* (%)	621 (4.7)	10 (4.8)	5 (3.7)	1 (4.8)	0.960
Hypertension *n* (%)	6849 (50.5)	118 (57.0)	76 (56.7)	12 (57.1)	0.122
Dyslipidemia *n* (%)	5352 (39.5)	88 (42.5)	63 (47.0)	9 (42.9)	0.263
Diabetes mellitus *n* (%)	2585 (19.0)	48 (23.2)	28 (20.9)	3 (14.3)	0.413
Obesity (BMI > 30) *n* (%)	2617 (21.2)	58 (28.0)	32 (23.9)	4 (19.0)	0.102
Anxiety *n* (%)	1045 (7.7)	19 (9.2)	11 (8.2)	3 (14.3)	0.594
Depression *n* (%)	1375 (10.2)	30 (14.5)	19 (14.2)	3 (14.3)	0.079
Neurodegenerative disorder *n* (%)	1237 (9.1)	21 (10.1)	16 (11.9)	1 (4.8)	0.573
Dementia *n* (%)	1358 (10.0)	25 (12.1)	14 (10.4)	1 (4.8)	0.653
Hemiplegia/paraplegia *n* (%)	221 (1.6)	6 (2.9)	4 (3.0)	0 (0.0)	0.281
Dialysis *n* (%)	154 (1.1)	5 (2.4)	4 (3.0)	0 (0.0)	0.145
Organ transplantation *n* (%)					<0.001
- No transplant - Kidney - Liver - Heart - Lung	13,246 (98.8) 116 (0.8) 23 (0.2) 11 (0.1) 14 (0.1)	202 (97.6) 5 (2.4) 0 (0.0) 0 (0.0) 0 (0.0)	131 (97.8) 2 (1.5) 1 (0.7) 0 (0.0) 0 (0.0)	20 (95.2) 0 (0.0) 0 (0.0) 0 (0.0) 1 (4.8)	
Arteriosclerotic vascular disease *n* (%)	2359 (17.4)	48 (23.2)	29 (21.6)	1 (4.8)	0.033
Non-arteriosclerotic vascular disease *n* (%)	1995 (14.7)	46 (22.2)	28 (21.0)	4 (19.0)	0.004
Respiratory pathology *n* (%)	2682 (19.7)	70 (33.8)	38 (28.3)	7 (33.3)	<0.001
HIV *n* (%)	96 (0.7)	0 (0.0)	0 (0.0)	0 (0.0)	0.460
Chronic liver disease, moderate to severe *n* (%)	133 (1.0)	3 (1.4)	2 (1.5)	0 (0.0)	0.799
Chronic kidney failure moderate-severe *n* (%)	804 (5.9)	21 (10.1)	11 (8.2)	1 (4.8)	0.055
Cancer *n* (%)	1425 (10.5)	24 (11.6)	12 (8.9)	2 (9.5)	0.891
Gastroduodenal ulcer *n* (%)	340 (2.5)	9 (4.3)	6 (4.5)	0 (0.0)	0.148
Treatment before admission					
Antimalarials *n* (%)	38 (0.3)	29 (14.0)	3 (2.2)	0 (0)	<0.001
Corticosteroids *n* (%)	428 (3.1)	105 (50.7)	59 (44.0)	7 (33.3)	<0.001
cs/tsDMARDs *n* (%)	360 (2.7)	97 (46.8)	28 (20.9)	8 (38.1)	<0.001
bDMARDs *n* (%)	149 (1.1)	21 (10.1)	10 (7.5)	1 (4.8)	<0.001

Data for patients with ADs were imputed for sociodemographic variables, comorbidity, and treatment before admission. Arteriosclerotic vascular disease (ischemic heart disease, cerebral vascular disease, and peripheral arterial disease). Non-arteriosclerotic vascular disease (heart failure and atrial fibrillation). Respiratory pathology (asthma, COPD, chronic bronchitis, sleep apnea syndrome). AD: autoimmune disease; bDMARDs: disease-modifying anti-rheumatic drugs, original biological or similar; BMI: body mass index; COPD: chronic obstructive pulmonary disease; cs/tsDMARDs: disease-modifying anti-rheumatic drugs, synthetic, conventional, or targeted; HIV: human immunodeficiency virus.

**Table 3 jcm-10-01844-t003:** Immunosuppressors/immunomodulators during the admission.

	Without AD	Classical AD	Other AD	Miscellaneous AD	*p*-Value
Hydroxychloroquine *n* (%)	11,581 (85.6)	172 (83.1)	106 (79.1)	19 (90.5)	0.113
Chloroquine *n* (%)	622 (4.6)	13 (6.3)	9 (6.7)	0 (0.0)	0.310
Tocilizumab *n* (%)	1136 (8.4)	19 (9.2)	9 (6.7)	3 (14.3)	0.661
Immunoglobulin *n* (%)	58 (0.4)	5 (2.4)	1 (0.7)	0 (0.0)	0.001
Anakinra *n* (%)	75 (0.6)	2 (1.0)	0 (0.0)	0 (0.0)	0.685
Baricitinib *n* (%)	77 (0.6)	3 (1.4)	0 (0.0)	1 (4.8)	0.020
Corticosteroids *n* (%)	4656 (34.5)	102 (49.3)	49 (36.6)	11 (52.4)	<0.001

AD: autoimmune diseases.

**Table 4 jcm-10-01844-t004:** Outcomes.

	Without AD	Classical AD	Other AD	Miscellaneous AD	*p*-Value
Length of stay (days) mean ± SD	11.2 ± 10.0	12.2 ± 10.7	11.4 ± 13.2	12.3 ± 8.0	0.502
NIMV *n* (%)	666 (4.9)	15 (7.4)	2 (1.5)	1 (4.8)	0.118
IMV *n* (%)	882 (6.5)	13 (6.4)	6 (4.5)	2 (9.5)	0.753
ICU admission *n* (%)	1108 (8.2)	23 (11.1)	7 (5.2)	2 (9.5)	0.264
Complications during admission *n* (%)	6246 (46.0)	118 (57.0)	70 (52.2)	10 (47.6)	0.008
Bacterial pneumonia *n* (%)	1460 (10.8)	34 (16.4)	20 (14.9)	1 (4.8)	0.022
ARDS *n* (%)	4414 (32.8)	84 (40.6)	48 (35.8)	7 (33.3)	0.019
Heart failure *n* (%)	765 (5.7)	23 (11.1)	10 (7.5)	0 (0.0)	0.004
Arrhythmia *n* (%)	523 (3.9)	18 (8.7)	7 (5.2)	1 (4.8)	0.005
Myocardial infarction *n* (%)	108 (0.8)	3 (1.4)	2 (1.5)	0 (0.0)	0.572
Myocarditis *n* (%)	118 (0.9)	4 (1.9)	2 (1.5)	0 (0.0)	0.346
Seizures *n* (%)	80 (0.6)	3 (1.5)	0 (0.0)	0 (0.0)	0.326
Stroke *n* (%)					
- Ischemic - Hemorrhagic	81 (0.6) 11 (0.1)	0 (0.0) 0 (0.0)	3 (2.2) 0 (0.0)	0 (0.0) 0 (0.0)	0.267
Shock *n* (%)	602 (4.5)	13 (6.3)	9 (6.7)	0 (0.0)	0.253
Acute kidney failure *n* (%)	1872 (13.9)	42 (20.3)	20 (14.9)	5 (23.8)	0.032
Sepsis *n* (%)	822 (6.1)	18 (8.7)	14 (10.4)	1 (4.8)	0.081
DIC *n* (%)	148 (1.1)	4 (1.9)	2 (1.5)	0 (0.0)	0.636
Thromboembolic disease *n* (%)					
- DVT - PE - DVT + PE	66 (0.5) 195 (1.4) 15 (0.1)	2 (1.0) 6 (2.9) 0 (0.0)	1 (0.7) 2 (1.5) 0 (0.0)	0 (0.0) 1 (4.8) 0 (0.0)	<0.001
Multiorgan failure *n* (%)	831 (6.2)	22 (10.6)	10 (7.5)	0 (0.0)	0.033
Peripheral arteriopathy *n* (%)	72 (0.5)	0 (0.0)	1 (0.7)	0 (0.0)	0.718
Readmission *n* (%)	496 (3.8)	11 (5.5)	5 (3.8)	0 (0.0)	0.495
Related to COVID-19 *n* (%)	205 (41.5)	7 (63.6)	0 (0.0)	0 (0.0)	0.056
Death *n* (%)	2968 (22.2)	64 (30.9)	49 (36.6)	2 (9.5)	<0.001
Composite outcome *n* (%)	3790 (27.9)	80 (38.6)	52 (38.8)	4 (19.0)	<0.001

AD: autoimmune disease; ARDS: acute respiratory distress syndrome; DIC: disseminated intravascular coagulopathy; DVT: deep venous thrombosis; ICU: intensive care unit; IMV: invasive mechanical ventilation; NIMV: non-invasive mechanical ventilation; PE: pulmonary embolism.

**Table 5 jcm-10-01844-t005:** Measures of the effect, OR (95% CI), of the ADs, sociodemographic characteristics, comorbidity, and immunomodulatory treatments prior to admission and during hospital stay in relation to the outcome variables ^α^.

Outcome	Mortality	Composite Outcome	Complications
*n* (%)	2449 (21.7)	3176 (27.7)	5309 (46.4)
AUC	AUC = 0.83 (95% CI 0.82–0.84)	AUC = 0.74 (95% CI 0.73–0.75)	AUC = 0.71 (95% CI 0.70–0.72)
Age, years	1.08 (1.08–1.09)	1.05 (1.04–1.05)	1.04 (1.03–1.04)
Sex (female)	0.58 (0.52–0.66)	0.60 (0.55–0.66)	0.60 (0.55–0.65)
Race			
- Caucasian - Black - Hispanic - Asian - Others	NS	1 1.06 (0.49–2.27) 1.43 (1.18–1.74) 1.19 (0.55–2.57) 1.15 (0.68–1.96)	1 0.87 (0.46–1.63) 1.45 (1.24–1.69) 0.77 (0.40–1.47) 1.06 (0.68–1.63)
Degree of dependence (Barthel Index score)			
- Absent/mild - Moderate - Severe	1 1.69 (1.45–1.98) 2.23 (1.87–2.66)	1 1.55 (1.34–1.79) 1.96 (1.66–2.31)	1 1.35 (1.16–1.57) 1.51 (1.27–1.79)
Hypertension	1.13 (1.01–1.27)	1.11 (1.01–1.23)	1.20 (1.10–1.32)
Dyslipidemia	NS	NS	1.09 (1.00–1.19)
Obesity (BMI > 30)	1.33 (1.17–1.51)	1.33 (1.19–1.48)	1.41 (1.28–1.55)
Respiratory pathology	NS	NS	1.14 (1.03–1.26)
Non-arteriosclerotic cardiovascular disease	NS	1.15 (1.02–1.30)	1.31 (1.16–1.48)
Habitual treatment with CS	1.81 (1.44–2.27)	1.54 (1.26–1.88)	1.39 (1.13–1.71)
Organ transplantation			
- No transplant - Kidney - Liver - Heart - Lung	1 1.83 (1.14–2.94) 1.78 (0.68–4.68) 0.78 (0.09–7.07) 10.32 (3.00–35.51)	NS	1 1.63 (1.06–2.51) 1.14 (0.48–2.72) 1.57 (0.39–6.31) 6.42 (1.37–30.16)
Charlson Comorbidity Index	1.16 (1.13–1.19)	1.12 (1.09–1.15)	1.08 (1.05–1.11)
Autoimmune disease	NS ^α,β^	NS ^α,β^	NS ^α,β^

^α^ Obtained in the logistic regression analysis (selection of variables: Wald statistician forward, with inclusion *p* < 0.05 and exclusion *p* < 0.10) in relation to the result variables: mortality, composite outcome (mortality, mechanical ventilation, and ICU admission), and complications during hospitalization. Predictive variables included in the model: absent/present AD, age, sex, race, alcoholism, smoking, Barthel Index score, hypertension, dyslipidemia, diabetes mellitus, obesity, depression, neurodegenerative pathology, dementia, moderate to severe renal pathology, dialysis (absent/hemodialysis/peritoneal), organ transplantation, respiratory pathology (asthma, COPD, chronic bronchitis, sleep apnea syndrome), non-arteriosclerotic cardiovascular disease (heart failure or AF); arteriosclerotic cardiovascular disease (TIA, stroke, hemiplegia, angina, AMI, peripheral arterial disease), severe liver disease, any neoplasm, gastro-duodenal ulcer disease, AIDS, HIV infection, Charlson Comorbidity Index, in-hospital complications, pre-admission immunosuppressive therapy with CS, antimalarials, bDMARDs, and cs/tsDMARDs. In total, there were 31 exposure variables. ^β^ Statistical significance was not reached in the final model for the AD variable categorized into four groups (absent, classic AD, other AD, and miscellaneous), nor when the immunoregulatory treatments used during hospitalization were included in the model (HCQ, CQ, tocilizumab, IGIV, anakinra, baricitinib, CS). AD: autoimmune diseases; AF: atrial fibrillation; AIDS: acquired immune deficiency syndrome; AMI: acute myocardial infarction; AUC: area under the ROC curve; bDMARDs: disease-modifying anti-rheumatic drugs, original, biological, or similar; BMI: body mass index; CQ: chloroquine; CS: corticosteroids; cs/tsDMARDs: disease-modifying anti-rheumatic drugs, synthetic, conventional, or targeted; CVA: cerebrovascular accident; HCQ: hydroxychloroquine; HIV: human immunodeficiency virus; ICU: intensive care unit; IVIG: intravenous immunoglobulins; NS: not significant; OSAS: obstructive sleep apnea syndrome; TIA: transient ischemic attack.

**Table 6 jcm-10-01844-t006:** Measures of the effect, OR (95% CI) *, of ADs as a whole and of immunomodulatory treatments prior to hospital admission with their propensity scores on the outcome variables ^α^.

		Antimalarials	Corticosteroids	cs/tsDMARDs	bDMARDs
Mortality	2.11 (0.56–7.58)	1.02 (0.29–3.58)	0.80 (0.26–2.49)	1.31 (0.60–2.90)	0.14 (0.03–0.78) *
Composite outcome	1.58 (0.44–5.63)	1.18 (0.38–3.68)	0.80 (0.28–2.33)	1.35 (0.65–2.80)	0.22 (0.04–1.14)
Complications	0.94 (0.25–3.44)	1.76 (0.56–5.55)	1.57 (0.47–5.30)	0.98 (0.47–2.02)	0.71 (0.09–5.91)

^α^ Propensity scores were obtained with the inverse weighting method. Outcome variables were mortality, composite outcome (mortality, mechanical ventilation, and ICU admission), and complications during the hospitalization. Predictor variables included in the logistic regression model to estimate the propensity score: age, sex, race, form of acquisition (common/nosocomial/residency), alcohol use disorder, smoking, Barthel Index score, comorbidity, Charlson Comorbidity Index, immunosuppressive/immunomodulatory treatments prior to admission (anti-malarial, glucocorticoids, cs/tsDMARDs, bDMARDs) and during hospitalization (antimalarials, tocilizumab, immunoglobulins, anakinra, baricitinib, corticosteroids), and the presence of a complication during hospitalization (38 exposure variables to calculate the propensity scores). A total of 11,174 patients hospitalized by COVID-19 were included. AD: autoimmune disease; bDMARDs: disease modifying anti-rheumatic drugs, original, biologic, or similar; cs/tsDMARDs: disease-modifying anti-rheumatic drugs, synthetic, conventional, or targeted; ICU: intensive care unit.

## Data Availability

The data this study is based on are available from the corresponding author upon reasonable request.

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
