# Peer review of "Autoimmune Diseases and COVID-19 as Risk Factors for Poor Outcomes: Data on 13,940 Hospitalized Patients from the Spanish Nationwide SEMI-COVID-19 Registry"

_jcm, 2021, doi:10.3390/jcm10091844_

Round 1

Reviewer 1 Report

Thank you for sending us the manuscript entitled: ²Autoimmune diseases and COVID-19 as risk factors for poor outcome. Data on 13,940 hospitalized patients from the Spanish National Registry SEMI-COVID-19² for reviewing.

There are several issues which need to be clarified.

  1. How prevalent is SARS-CoV-2 infection among patients with autoimmune diseases (AD)? Does the treatment with cs or/and bDMARDs impact the disease course and outcome? In other words, from the SEMI registry how many patients with AD receiving cs and/or bDMARDs have been affected from SARS-CoV-2 infection and were not hospitalized. What is the percentage of the hospitalized patients with AD?
  2. The author have included in the present study patients with AD like PMR, sarcoidosis, idiopathic pulmonary fibrosis, erythema modicum and others, which really are not classified as autoimmune diseases. These are classified as chronic inflammatory disorders.
  3. In Table 1. The authors present many AD like RA , PSS, SLE, scl etc. but they analyzed them as an entity. The AD are very heterogeneous disorders, ranging from a mild disease like PSS, to very severe diseases like scleroderma, SLE etc. In these disorders the treatment is also different. For PSS the treatment did not require cs or/and bDMARDs while in SLE high doses of steroids and many immunosuppressive drugs are used.
  4. In Table 1. The authors present 23 patients with SLE and 13 patients with scleroderma. In Table 2. they present the patients’ baseline characteristics and comorbidities, which comprise cancer, pulmonary disorders transplant patients etc. In the bottom on this table, the authors describe the drugs cs, ts and bDMARDs, as well as steroids and antimalarial receiving from their patients before admission. The questions which arise here is, no other drugs have been receiving from patients with cancer, pulmonary disorders and transplant patients?

There are several reports in the literature dealing with the same subjects which demonstrated that the chronic use of cs and/or bDMARDs may be having a beneficial effect of SARS-CoV-2 infection in these patients. It seem that COVID-19 disease in AD patients have a mild course and no severe consequences in the majority of them. For example:

Ann Rheum Dis 2020,79:667-668

Arhritis Rheumatol 2020,72:1981-1989

Rheumatol Int 2021,doi.org/10.1007/500296-021-04818-2

Nat Rev Rheumatol 2021,  17: 71-72

Clin Exp Rheumatol 2020, 38:175-180

Mediterr J. Rheumatol 2020,31(suppl 2): 259-267

Ann Rheuma Dis 2021, doi:101136/annrheumadis-2020219498

Reviewer 2 Report

In the present study, Gutiérrez et al. examine the clinical characteristics and the clinical course of hospitalized patients with COVID-19 and autoimmune diseases (AD) compared to the general population. The study is well performed and presented, especially relevant in AD patients under biological disease-modifying anti-rheumatic drugs (b-DMARD) with concomitant COVID-19. I have minor suggestions to improve the quality of the study:

  1. Hyperinflammatory syndromes should be more discussed in detail: cytokine release syndrome (CRS), immune reconstitution inflammatory syndrome (IRIS, e.g. doi: 10.1136/annrheumdis-2020-218836) especially with regard of b-DMARD therapy.
  2. It would be of interest if the serological response to SARS-CoV-2 infection was assessed.
  3. In addition, were also patients included that already were vaccinated?
  4. It would be of interest to perform subgroup analyses for outcome measures separating patients admitted to the ICU and non-ICU patients. 
  5. Minor: please correct the references throughout the manuscript text (point, comma behind the reference, correct space symbols). 

Round 2

Reviewer 1 Report

I no have further comments